



# Precipitation Biases and Snow Physics Limitations Drive the Uncertainties in Macroscale Modeled Snow Water Equivalent

Eunsang Cho[1,2], Carrie M. Vuyovich[1], Sujay V. Kumar[1], Melissa L. Wrzesien[1,2], Rhae Sung Kim[1,3], Jennifer M. Jacobs[4,5]

[1]Hydrological Sciences Laboratory, NASA Goddard Space Flight Center, Greenbelt, MD, USA
[2]Earth System Science Interdisciplinary Center, University of Maryland, College Park, MD, USA
[3]Goddard Earth Sciences Technology and Research II, University of Maryland Baltimore County, Baltimore, MD, USA
[4]Department of Civil and Environmental Engineering, University of New Hampshire, Durham, NH, USA
[5]Earth Systems Research Center, Institute for the Study of Earth, Oceans, and Space, University of New Hampshire, Durham, NH, USA

*Correspondence to*: Eunsang Cho (eunsang.cho@nasa.gov)

**Abstract.** Seasonal snow is an essential component of regional and global water and energy cycles, particularly in snow-dominant regions that rely on snowmelt for water resources. Land surface models (LSMs) are a common approach for developing spatially and temporally complete estimates of snow water equivalent (SWE) and hydrologic variables at a large scale. However, the accuracy of the LSM-based SWE outputs is limited and unclear by mixed factors such as uncertainties in the meteorological boundary conditions and the model physics. In this study, we assess the SWE, snowfall, precipitation, and air temperature products from a twelve-member ensemble - with four LSMs and three meteorological forcings - using automated SWE, precipitation, and temperature observations from 809 Snowpack Telemetry stations over the western U.S. Results show that the mean annual maximum LSM SWE is underestimated by 268 mm. The timing of peak SWE from the LSMs is on average 36 days earlier than that of the observations. By the date of peak SWE, winter accumulated precipitation is underestimated (forcings mean: 485 mm vs. stations: 690 mm). In addition, the precipitation partitioning physics generates different snowfall estimates by an average of 113 mm with the same forcing data. Even though there are widespread cold biases (up to 3℃) in the temperature forcings, larger ablations and lower ratios of SWE to total precipitation are found even in the accumulation period, indicating that melting physics in LSMs drives some SWE uncertainties. Based on the principal component analysis, we find that precipitation bias has the largest contribution to the first principal component, which accounts for more than half of the total variance. The results provide insights into prioritizing strategies to improve SWE estimates from LSMs for hydrologic applications.



# 1 Introduction

Seasonal snow plays a critical role in hydrologic and climatologic processes globally (Sturm et al., 2017). It benefits 1.2 billion of the world's population via its seasonal cycle which retains water for release during warm and dry periods (Barnett et al., 2005; Viviroli et al., 2007). Also, the impact of seasonal snowpack on extreme events such as snowmelt and rain-on-snow floods (Davenport et al., 2020; Li et al., 2019; Musselman et al., 2018), drought (Huning and AghaKouchak, 2020), and wildfires (Westerling et al., 2006) are of considerable interest to the water resources planners and decision-makers. Despite being a vital component of water balances and extreme events, estimating the spatiotemporal change in snow water storage referred to as snow water equivalent (SWE), remains a significant challenge for the snow hydrology community (Dozier et al., 2016; Kim et al., 2021).

Land surface models (LSM) are commonly used to quantify distributed estimates of SWE and other hydrologic variables with high spatiotemporal resolution over large spatial extents. Because LSMs enable the simulation of the physical interactions between water, energy, and carbon cycle processes on the land surface, they are widely used for hydrologic and climate research, as well as operational applications related to river flow forecasting and weather prediction (Mitchell et al., 2004), among others. However, recent studies revealed that there are large uncertainties in SWE estimates from LSMs (Broxton et al., 2016; Kim et al., 2021; Pan et al., 2003), which lead to uncertainties in other relevant variables and processes such as snowmelt runoff, spring soil moisture, and groundwater recharge. Broxton et al. (2016) found that the LSM-based reanalysis SWE products were largely underestimated as compared to the high-resolution reference SWE data sets and suggested that more rapid snow ablation in LSMs especially at near-freezing temperatures is the primary source of the SWE underestimates. In contrast, some studies found that biases in atmospheric forcing data, particularly precipitation, were a major driver of SWE error (Pan et al., 2003; Raleigh et al., 2015), and reanalysis products, often used as meteorological forcing for LSMs, underestimated precipitation, particularly in mountainous areas (Enzminger et al., 2019; Wrzesien et al., 2017). Typically, reanalysis products are limited in capturing the orographic effects near mountainous areas, subsequently resulting in underestimates of precipitation (Wrzesien et al., 2017).

Several snow model intercomparison exercises provided a series of valuable findings according to their objectives. Phase 1 and 2 of the Snow Model Intercomparison Project (SnowMIP) focused on snow energy-budget simulations (Etchevers et al., 2004) and forest snow processes (Essery et al., 2009; Rutter et al. 2009), respectively, based on site-scale reference simulations. SnowMIP1 found that, while the complex snow models were better able to simulate net longwave radiation, the model complexity had relatively little impact on albedo simulation based on two mountainous alpine sites (Etchevers et al., 2004). In SnowMIP2, Essery et al. (2009) found the snow models generally captured the large differences in albedo and surface temperature quite well between snow-covered and snow-free surfaces and between forested and open sites at three open and forested site pairs. They also concluded that there was little consistency in model performance among sites and years, so no "best" model was identified. Recently, the Land Surface, Snow and Soil moisture MIP (LS3MIP v1.0; van den Hurk et al., 2016) and Earth system models-SnowMIP (ESM-SnowMIP; Krinner et al., 2018 and Menard et al., 2021), an extension of



LS3MIP, focused on assessing snow models and snow–climate interactions at local and global scales. Krinner et al. (2018) found that albedo simulation was a major source of uncertainty in the context of snow-related climate feedback based on ten site experiments for a local-scale assessment. However, these model intercomparison projects did not evaluate SWE at macro-scales needed for operational water resources management. Krinner et al. (2018) and these project exercises suggest that longer periods and larger study domains (and more sites over various snow environments) would improve our understanding of snow

models and land surface schemes in Earth system models and reduce the uncertainties associated with snow processes and feedbacks on the climate.

While the previous studies of continental (or global) scale SWE evaluations outline sources of the SWE errors (Broxton et al., 2016; Kim et al., 2021; Pan et al., 2003; Mortimer et al., 2020), comprehensive quantification of the relative contribution of these error sources is still required. Furthermore, most of the prior studies used a single or multiple LSMs with a single

meteorological forcing and/or simulated/reanalysis SWE with relatively coarse spatial resolutions (e.g., 12.5 km to 50 km), which impedes the quantification of the contributions by producing additional uncertainties. To overcome this, we examine data from a multi-LSM, multi-forcing ensemble at 5-km spatial resolution (called the Snow Ensemble Uncertainty Project; SEUP) developed by Kim et al. (2021), which includes twelve combinations of 5 km-resolution gridded SWE products from four LSMs and three meteorological forcings. Kim et al. (2021) quantified the spatial and temporal uncertainties in SWE and

total snow storage uncertainty across North America (They referred to "uncertainty" as the range of SWE estimates across the twelve ensemble members). The largest uncertainty in the modeled SWE was found in mountainous regions. They speculated that this was due to the relative deep snow, meteorological forcing uncertainties, and variability among the different snow physics in the LSMs over complex terrain. Also, they demonstrated that SWE uncertainty drives runoff uncertainty, suggesting that improved SWE observations are required to reduce the SWE and runoff uncertainty, particularly during the melt season

in high-latitude regions (e.g., northern Canada) and the western mountain regions.

This study seeks to identify the primary sources of the errors and to quantify their contributions to the modeled SWE uncertainty (forcing errors vs. snow-related physics) during the accumulation periods for eight years (2010 to 2017) against the 809 Snowpack Telemetry (SNOTEL) stations. We focus on the snow accumulation period which is defined as October 1st to the date of the annual maximum SWE of each station. Previous studies have investigated LSM uncertainty during the

90 melting season, such as too rapid snowmelt (e.g. Broxton et al., 2016), though less work has considered the accumulation season. We aim to answer the following questions - (1) How large is the LSM SWE uncertainty as compared to SNOTEL measurements? (2) How much does underestimation in precipitation forcing contribute to snow uncertainty? (3) Do precipitation partitioning methods contribute to the SWE underestimation? (4) How much does LSM melt physics contribute to the SWE underestimation? and (5) What are the relative contributions among the sources of the error?



## 2 Data and methods

### 2.1 Snow Ensemble Uncertainty Project (SEUP)

The SEUP ensemble developed by Kim et al. (2021) is comprised of 12 ensemble members, created by the combination of four different LSMs – Noah with Multi-Parameterization version 3.6 (hereafter Noah-MP; Niu et al., 2011), Catchment version 2.5 (hereafter Catchment; Koster et al., 2000), Joint UK Land Environment Simulator (JULES; Best et al., 2011), and Noah version 2.7.1 (hereafter Noah; Ek et al., 2003) – and three different forcing datasets – European Centre for Medium-Range Weather Forecasts (ECMWF; Molteni et al., 1996), Global Data Assimilation System (GDAS; Derber et al., 1991), and Modern-Era Retrospective Analysis for Research and Applications, version 2 (MERRA2; Gelaro et al., 2017). These models are selected to provide a baseline of operational LSM capabilities for SWE estimation because they are used for operational purposes at major modeling centers such as the U.S. National Centers for Environmental Prediction, NASA Global Modeling and Assimilation Office, and the United Kingdom Met Office (Note that the model versions and detailed configurations used in this study could differ from what the centers are currently using). In the SEUP analysis, 3-hourly SWE estimates were generated for the 12 combinations using the NASA Land Information System (LIS; Kumar et al., 2006). All models were run at a 5-km resolution from 2000 to 2017 at 15 min time steps. To achieve initial hydraulic and thermal equilibrium states for each run, the first 10 water years from 2000 to 2009 were used as a model spin-up period and the remaining eight water years from 2010 to 2017 were used for the evaluation in this study. Noah LSM uses a one-layer snow model to simulate SWE by calculating snowfall minus the sum of sublimation and snowmelt. More detailed descriptions of Noah's physics and development are presented by Ek et al. (2003) and Koren et al. (1999). Noah-MP LSM is the advanced Noah model with new multiple options for selected processes (Niu et al., 2011). Noah-MP includes up to three layers of snowpack, depending on snow depth. The snow scheme calculates snow compaction from the weight of overlying snow layers and melting metamorphism. The Catchment LSM includes a three-layer snowpack model that incorporates snow physics including densification, snowmelt, refreeze, and snow insulating properties (Lynch-Stieglitz, 1994). The prognostic state variables for each layer include snow depth, snow cold content, and SWE. JULES LSM is run in stand-alone mode with a multilayer snow scheme driven by forcing data (Best et al., 2011). In the multilayer snow scheme, three layers of snowpack are set. For each layer, snow density is dynamically calculated by snow temperature over time, and thermal conductivity is calculated by using snow density. For further details, refer to Section 3 in the JULES model description paper (Best et al., 2011). To improve the spatial representativeness of the coarse resolution forcing inputs, the precipitation forcings were downscaled with the WorldClim monthly precipitation climatology (Fick and Hijmans, 2017). Other forcing variables including near-surface temperature were downscaled to 5 km by applying a constant lapse rate of 6.5 K km$^{-1}$, hypsometric adjustments using the Shuttle Radar Topography Mission (5 km; SRTM), and the USGS Global 30 Arc-Second Elevation (GTOPO30) data sets (Kim et al., 2021).

In the SEUP ensemble, two different methods are used for partitioning precipitation into rain or snow, (1) a single threshold method is used in the Catchment, JULES, and Noah, and (2) Jordan's fractioning method, which is used in Noah-MP (Jordan,





1991). A single threshold approach simply uses $T_{air}$ to determine the precipitation phase (Motoyama, 1990). Snowfall occurs whenever there is nonzero precipitation, and the near-surface air temperature is less than 0°C. The Jordan's method assumes that any precipitation is rainfall when $T_{air}$ > 2.5 °C, snowfall when $T_{air}$ < 0.5 °C, a snowfall fraction ($F_{snow}$) of 0.6 when 2.0 °C < $T_{air}$ ≤ 2.5 °C, and a linear equation (i.e. $F_{snow}$ = 1 - 0.2·Tair) from rainfall to snowfall between 0.5 °C < $T_{air}$ < 2.0 °C. Schematic diagrams of each method are provided in the Supporting Information (**Figure S1**). A detailed explanation of the SEUP framework can be found in Kim et al. (2021). For the comparison with daily SNOTEL observations in this study, we used the 3-hourly averaged SWE output at 00:00 Coordinated Universal Time (UTC). The air temperature ($T_{air}$) outputs from 00:00 and 12:00 UTC are averaged into a single daily average.

## 2.2 Snow Telemetry (SNOTEL)

For reference data sets, we use daily SWE observations measured at the 809 SNOTEL stations across the western United States operated by the Natural Resources Conservation Service (NRCS). These stations include accumulated precipitation and daily mean temperature observations co-located with the snow pillows and were used to evaluate the forcing precipitation and temperature data sets. Previous studies have identified a warm bias ranging from +0.5 to 2.0 °C at cold temperatures (less than 10 °C) in the SNOTEL temperature data (Figure S4 in Oyler et al., 2015). This bias is found to be temperature-dependent such that positive biases occur at colder temperatures (less than 12 °C) and negative biases occur at warmer temperatures (above 12 °C). In winter, the median biases for daily maximum and minimum observations were +1.25 °C and +1.75 °C. For this work, we used a bias-corrected temperature applied by a linear equation used in previous studies (Currier et al., 2017; Sun et al., 2019).

$$T_{corr} = 1.03 \cdot T_{ori} - 0.9 \tag{1}$$

where $T_{corr}$ is the bias-corrected temperature (°C) and $T_{ori}$ is the originally observed temperature (°C). While we acknowledge that the difference in spatial representativeness (point vs. grid) may lead to some uncertainties in the comparison, this topic is out of scope in this study. We assume that the SNOTEL stations are spatially representative around the areas, and more than 800 SNOTEL sites across the western U.S. would be sufficient to quantify macroscale LSM SWE uncertainties.

## 2.3 Principal Component Analysis

Principal Component Analysis (PCA) is a multivariate technique that is widely used to analyze a data table containing several variables that are generally dependent and inter-correlated (Abdi & Williams, 2010). It extracts important information about relationships among the variables and expresses them as a set of new orthogonal variables, called principal components (PCs). The PC loading measures the correlation between the PC and variables. Therefore, variables with similar loadings for a given PC can be dependent on the same common factor and are positively correlated. The goal of using PCA in this study is to identify the spatial similarity of the SWE difference between LSMs and observations to that of the potential error sources. In





our case, the data matrix arranges the SWE errors, *SWE_bias*, and four potential error sources including winter total
precipitation bias, *P_bias*, precipitation difference between two precipitation partitioning methods, *P_phase*, snow ablation

difference for the accumulation period as a proxy of melting physics, *Ablation*, and mean bias in winter air temperature, *T_bias*,
in columns and the 809 SNOTEL sites in rows. The "SWE_bias" is calculated by the ensemble mean SWE subtracted from
the corresponding SNOTEL SWE for each station. The potential sources of the error are obtained from the comparison between
SEUP and SNOTEL observations. The data matrix was pre-processed: the values in each column were normalized with the
following two steps: 1) the mean of each column is zero, and 2) each column was standardized to the unit norm as the variables

have different units.

## 3 Results

### 3.1 How large is the LSM's SWE uncertainty?

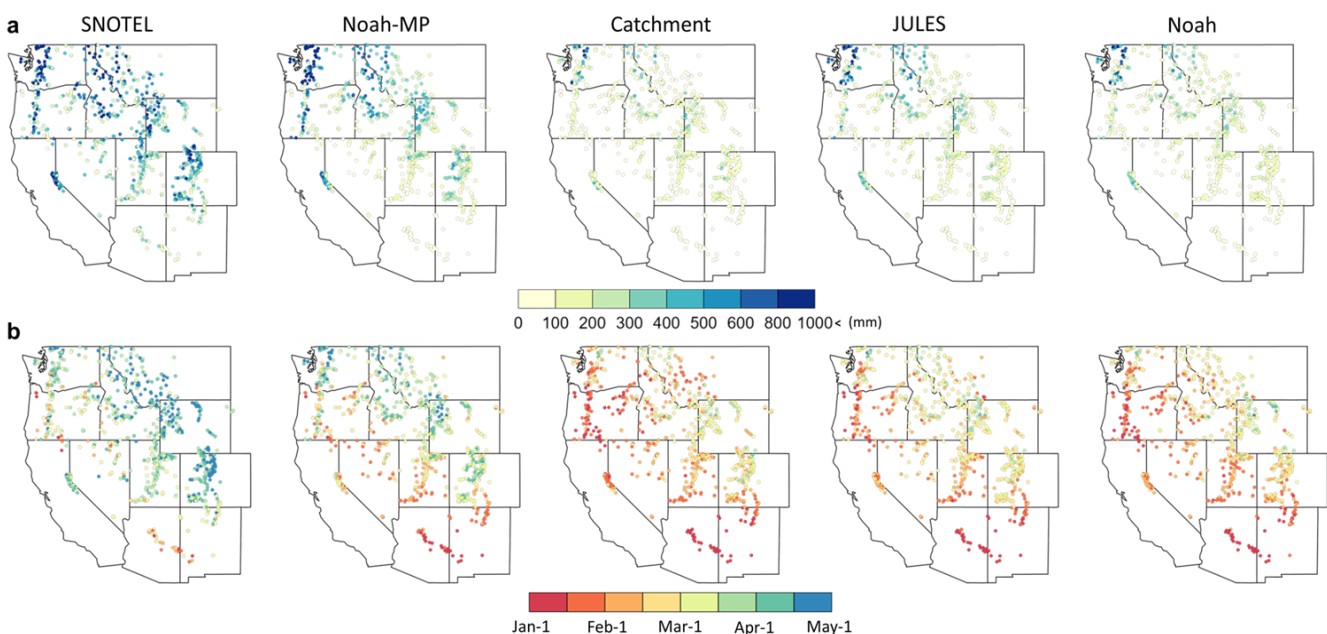

**Figure 1**. (a) Mean annual maximum SWE maps of SNOTEL observations and the four LSMs averaged over the three forcings
and (b) mean date maps when the annual maximum SWE occurs from 2010 to 2017 across the western United States

To quantify differences between LSM SWE estimates and ground observations, the mean magnitude and timing of annual

maximum SWE estimates for the four LSMs are compared to the corresponding SNOTEL SWE measurements across the
western United States from October 2009 to May 2017 (**Figure 1**; The same maps but for annual mean and April-1$^{st}$ SWE
difference are provided in **Figures S2 & S3**). There is a widespread underestimation of the annual maximum SWE for all
LSMs with averages of -198, -311, -273, and -288 mm for Noah-MP, Catchment, JULES, and Noah, respectively, although

the amount of the underestimation is regionally dependent on elevation ranges (**Figure 2b**). The mean timings of the annual

maximum LSM's SWE are on average 36 days earlier than the observation (21, 45, 38, and 42 days for Noah-MP, Catchment, JULES, and Noah, respectively). Larger underestimates occur in mountainous areas with higher elevation ranges such as the Pacific Northwest, the Sierra Nevada, and the Rockies, where larger SWE generally occurs (see the elevation map of **Figure S4**). Noah-MP provides relatively smaller differences as compared to the other three LSMs, though in some Pacific Northwest stations where there is more winter precipitation, Noah-MP SWE exceeds the SNOTEL SWE. Of the four LSMs, the snow-

related physics in Noah-MP generates SWE values most similar to observations.

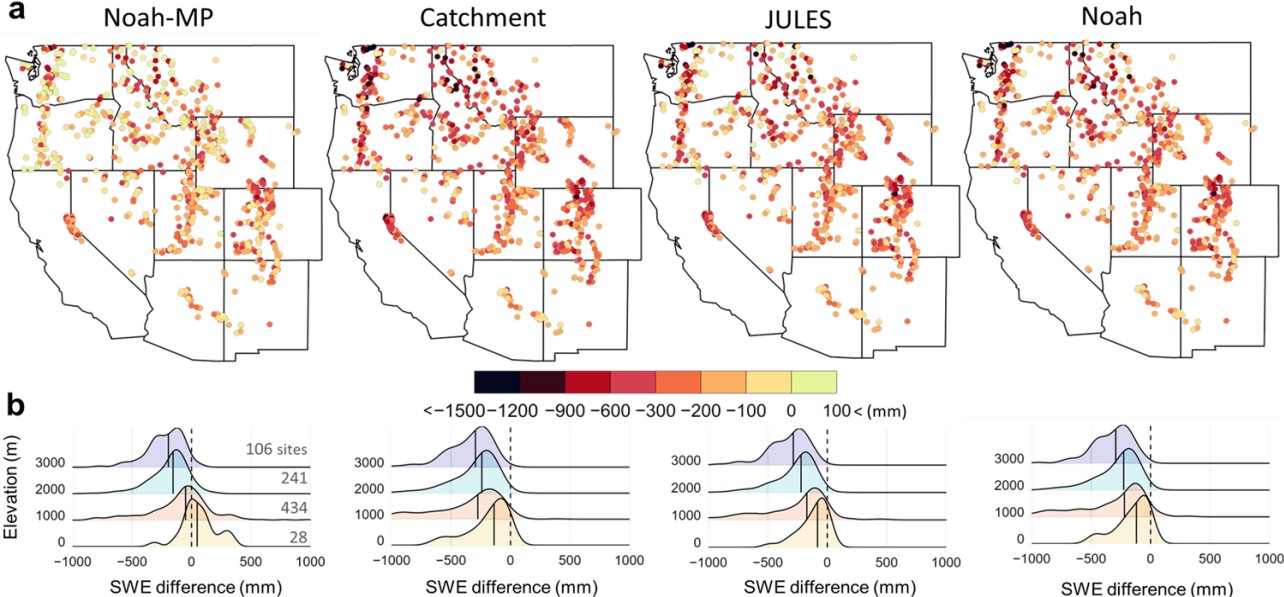

**Figure 2**. (a) Mean difference maps (four LSMs ensembled by three forcings *minus* SNOTEL) in the annual maximum SWE from 2010 to 2017 across the western United States and (b) density plots of the SWE difference by four elevation ranges of SNOTEL sites

### 3.2 Is precipitation forcing data underestimated?

As compared to the SNOTEL observations, ECMWF, GDAS, and MERRA2 forcings have widespread underestimates of winter accumulated precipitation (which is from Oct-1st to the date of the maximum SWE of each ensemble member for each station) in 98, 77, and 85% of the total stations, respectively. Of the stations, average biases (forcing *minus* SNOTEL) are -

356, -262, and -292 mm (maximum biases: -1432, -1523, and -1466 mm) for ECMWF, GDAS, and MERRA2, respectively (**Figure 3**). The largest underestimates were found in the Pacific Northwest, the Sierra Nevada, and Southern Rockies, particularly areas with higher elevation ranges (> 3000 m a.s.l.; **Figure 3b**). Among the three forcing data sets, GDAS provides a relatively lower overall difference as compared to ECMWF and MERRA2. In some stations in Washington and Wyoming, GDAS precipitation exceeds the SNOTEL observations. In stations with lower elevation ranges (0 – 1000 m a.s.l.), dominantly

located in the Pacific Northwest regions, there is a wider range of the precipitation difference between the forcings and
SNOTEL as compared to higher elevation ranges (**Figure 3b**).

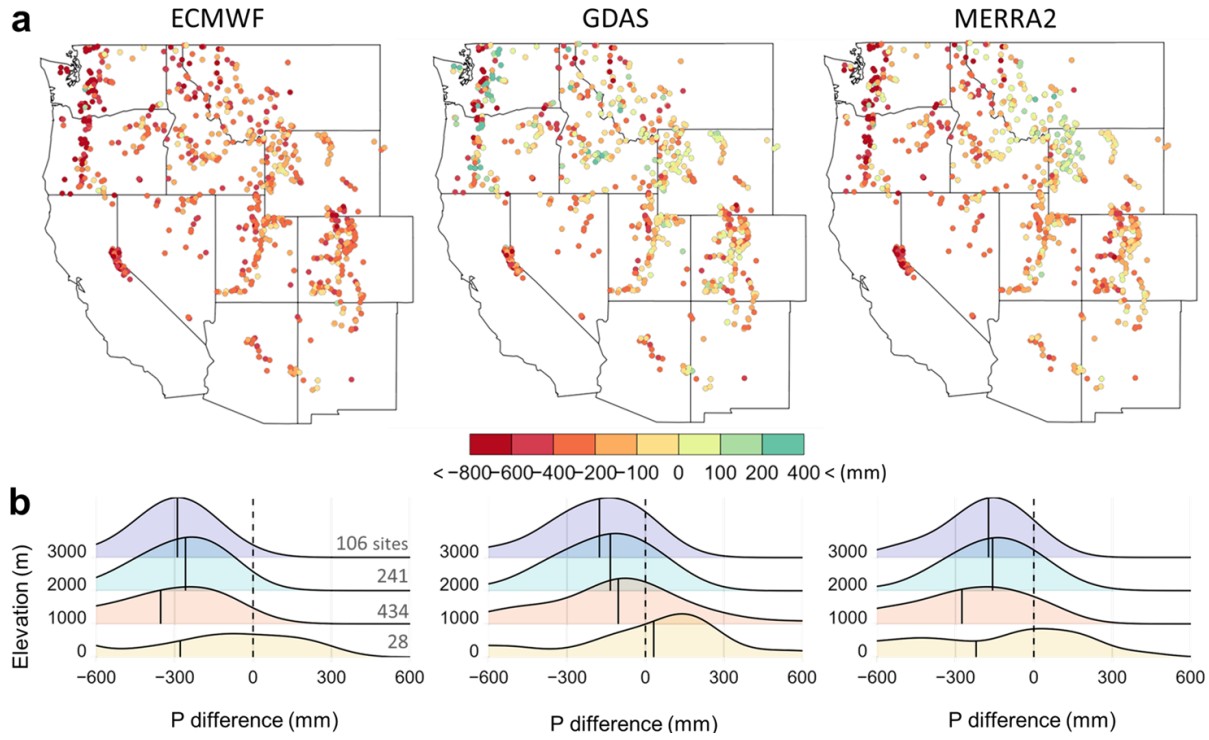

**Figure 3**. (a) Mean difference maps (three forcings ensembled by four LSMs minus SNOTEL) in winter accumulated
precipitation by the date of the maximum SWE of each ensemble member from 2010 to 2017 across the western United States
and (b) density plots of the precipitation difference by four elevation ranges of SNOTEL sites.

### 3.3 Does precipitation partitioning (rain vs. snowfall) contribute to the SWE underestimation?

The two different precipitation partitioning methods used in the LSMs generate different amounts of snowfall by region.
**Figure 4** provides mean annual total snowfall maps of Jordan's fractioning method (Jordan, 1991; used in Noah-MP) and a
single threshold method (other three LSMs) and the difference map between the two. The annual mean difference between the
two partitioning methods from 2010 to 2017 is about 113 mm for the 809 stations (24% of the winter accumulated precipitation
from October 1 to the max SWE dates [472 mm]). The differences are regionally dependent on elevation. The spatial patterns
of the three difference maps are similar regardless of meteorological forcing sources (**Figure S5**). This is not surprising because
the fractioning method partitions partial precipitation amounts with air temperatures ranging from 0 to 2.5 °C as snowfall,
which would be classified as liquid rainfall with a single threshold method that uses 0 °C as the rain-snow threshold. The

patterns of the larger snowfall are apparent in the Pacific Northwest where more precipitation occurs, and temperatures are frequently close to 0 °C in winter, as compared to other regions.

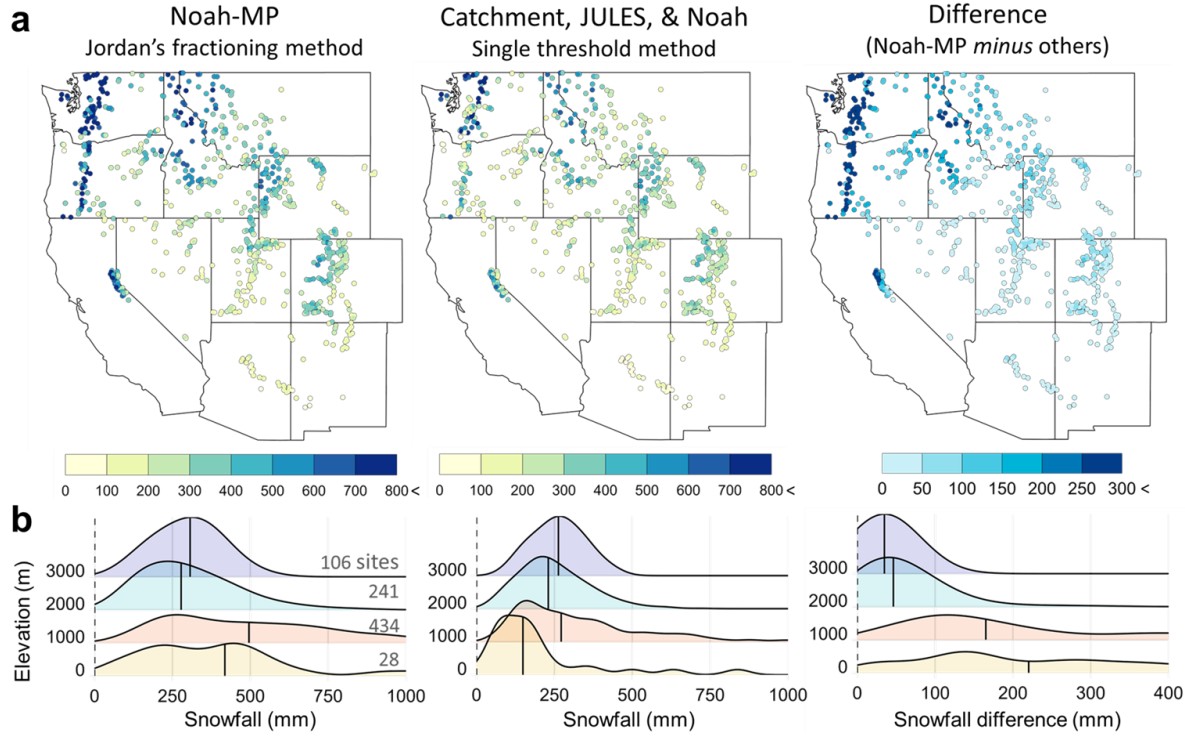

**Figure 4**. (a) Mean annual total snowfall maps during the accumulation period (Oct 1ˢᵗ to the date of the maximum SWE of each ensemble member) from 2010 to 2017 using Jordan (1991)'s fractioning method in Noah-MP and a single threshold method (0°C) in other three LSMs as well as the difference map across the western United States and (b) density plots of the precipitation difference by four elevation ranges of SNOTEL sites. The snowfall maps are ensembled by three forcings

The sensitivity of snowfall amounts due to the partitioning methods draws our attention to the role of air temperature differences among the three meteorological forcing datasets and the SNOTEL observations. In **Figure 5**, the mean annual temperatures from three forcing data sets are compared to the bias-corrected SNOTEL air temperature for the study period. In 68, 86, and 53% of stations, air temperatures from ECMWF, GDAS, and MERRA2 forcings exhibit negative (cold) biases as compared to the bias-corrected SNOTEL temperatures, respectively. Larger cold biases of ECMWF and GDAS are found in the continental regions with higher elevation ranges (e.g. -1.3 and -2.8 °C of the median biases for stations with > 3000 m a.s.l., respectively). MERRA2 has small biases for all elevation ranges even though there exist contrasting biases between continental (cold) versus maritime (warm) regions.

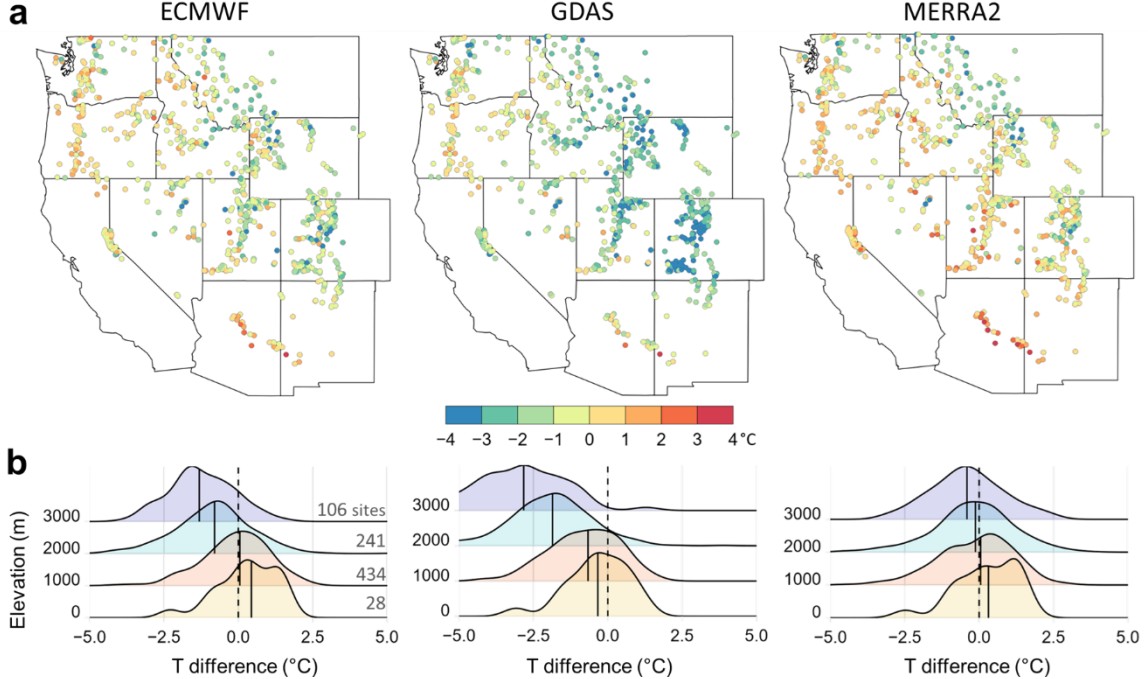

**Figure 5**. Mean differences in daily mean air temperature for winter period (1st October to 31st May) between three forcings (ECMWF, GDAS, and MERRA2) and bias-corrected SNOTEL data sets for eight water years from 2010 to 2017

### 3.4 Does melting physics in LSMs contribute to the SWE underestimation?

To examine the contribution of melting physics in LSMs to the SWE underestimation, the mean annual snow ablations during the accumulation period, i.e. from October 1st to the date of the maximum SWE for each year for each SEUP member, are compared to the corresponding amount of snow ablation from SNOTEL (**Figure 6**). The reason why the spring melt ablation period was excluded in the analysis is that it would include the impact of precipitation underestimation (i.e. the LSM ablation would always be lower than the SNOTEL ablation simply because there was less total snowfall to begin with). The same maps but for the accumulated snow ablation from October 1st to April 1st are provided in **Figure S6**. Even though the LSMs underestimate the SWE, and the peak SWE dates are generally earlier than the observations, there are larger ablations from all LSMs as compared to the observations at 79% to 89% of the total stations by up to 184 mm (Noah-MP; mean difference: 57 mm) to 274 mm (Catchment; mean difference: 57 mm), except for regions with lower elevation ranges (< 1000 m a.s.l.). The magnitude of the ablations is highly region-specific. While larger ablations are found in mountainous regions with higher elevation ranges (> 1000 m a.s.l.) such as the Northern Rockies and the Sierra Nevada, there is a tendency toward smaller ablations than the observations in the Pacific Northwest. In fact, this is because there was lower SWE accumulation from LSMs than observed, resulting in less snow ablation. Noah-MP has a relatively smaller difference in ablation as compared to the other three LSMs, suggesting that melting physics in Noah-MP probably performs better than that of others.





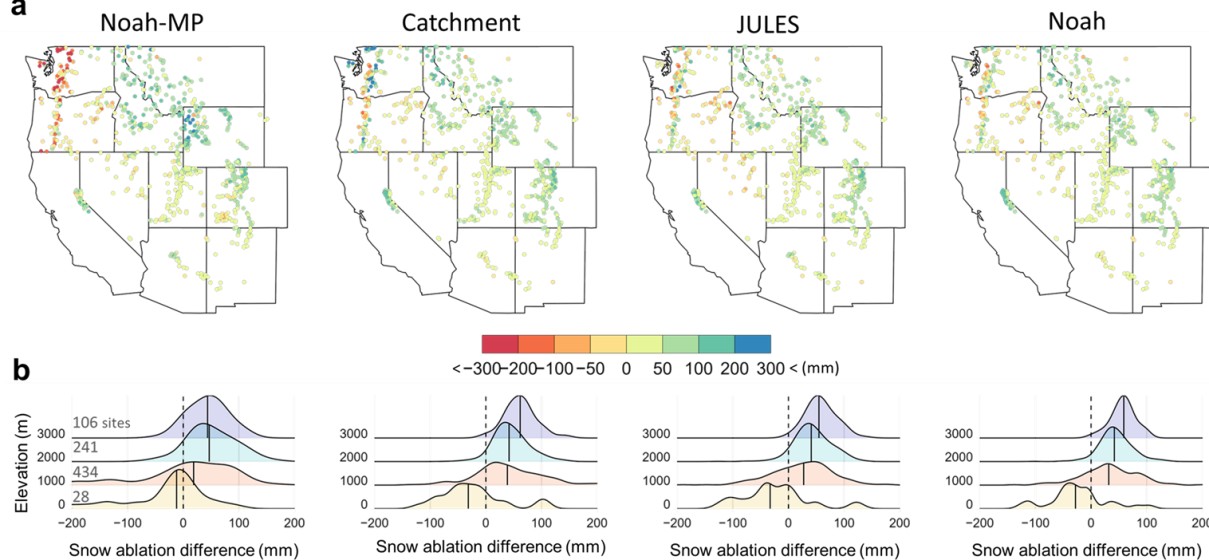

**Figure 6.** (a) Mean difference maps (four LSMs ensembled by three forcings *minus* SNOTEL) in the accumulated snow ablation during the snow accumulation periods from October 1 to the date of the maximum SWE of each ensemble member for each year from 2010 to 2017 across the western United States and (b) density plots of the snow ablation difference by four elevation ranges of SNOTEL sites

The ratios of SWE to total precipitation ($f_{SWE, precip}$) for each LSM and SNOTEL are also compared to examine the proportion of early snow ablation to precipitation (**Figure 7**). For the ratio map from SNOTEL, about 80 to 90% of the precipitation is accumulated as snowpack in the continental regions including the Sierra Nevada. Whereas SNOTEL has about 20 to 40% precipitation accumulated as SWE during the accumulation period in the Pacific Northwest and southern regions including Arizona and New Mexico. Results show that most LSMs underestimate the $f_{SWE, precip}$ and melting losses occur too frequently in LSMs during the accumulation period. Among the LSMs, the spatial distribution of the Noah-MP's ratio closely follows the SNOTEL observations, followed by Catchment, JULES, and Noah. However, Noah-MP still has lower ratios, particularly in the Middle/Northern Rockies as well as Arizona/New Mexico Mountains.





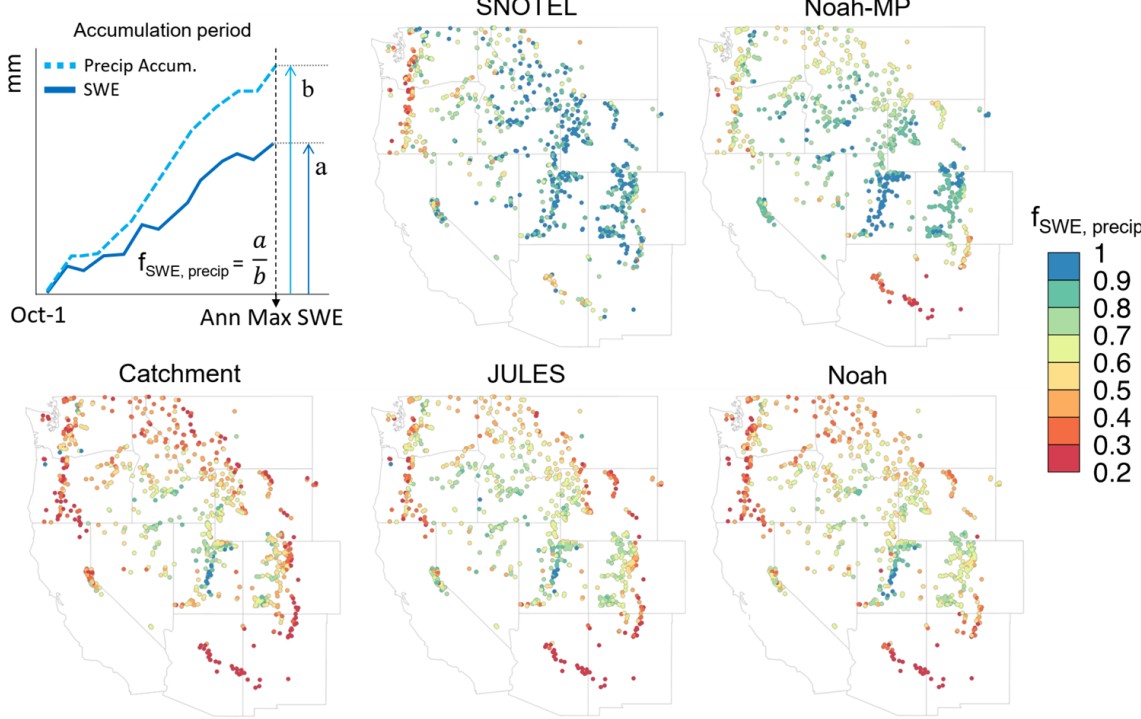

**Figure 7**. Maps of the mean ratio of the annual maximum SWE to total accumulated precipitation for the SNOTEL and four LSMs by the date of the maximum SWE of each ensemble member from 2010 to 2017. Each LSM result was ensembled by three forcings.

### 3.5 What is the relative contribution of potential causes on SWE uncertainties?

To quantify the relative contributions of the error sources on the SWE uncertainties, a principal component analysis (PCA) is conducted with the SWE biases (*SWE_bias*) and four identified error sources (*P_bias*, *P_phase*, *T_bias,* and *Ablation*) for the 809 stations (**Figure 8**). The PC loadings for the four prominent PCs jointly account for about 95% of the total spatial variance in the PCA data. With respect to the PCA with *SWE_bias* in **Figure 2**, the spatial variability in the SWE errors is prominently featured in the first three PCs that account for 85.7% of the total variance. The first PC (PC1; 50.7% explained) shows positive correlations between *SWE_bias* and *P_bias*, *P_phase*, and *Ablation*, except for *T_bias*. That is, areas of higher *P_bias*, *P_phase*, and *Ablation* are associated with larger SWE_bias, and these areas correspond to Northern Rocky Mountains. Among them, *P_bias* (0.55) has a larger contribution to PC1 than *P_phase* (0.47) and *Ablation* (0.46). The second PC (PC2; 17.6% explained) describes the positive correlations between *SWE_bias* and *T_bias*. PC2 indicates that there are areas where the SWE errors and temperature differences are positively correlated, and these areas correspond to Colorado (e.g., Southern Rocky Mountains) in **Figure 5**. Next, the third PC (PC3; 16.4% explained) shows negative correlations between *SWE_bias* and all error sources. That is, areas of smaller *P_bias*, positive *P_phase*, *T_bias*, and *Ablation* are associated with the SWE underestimation (negative), and the areas correspond to Arizona and New Mexico Mountains.

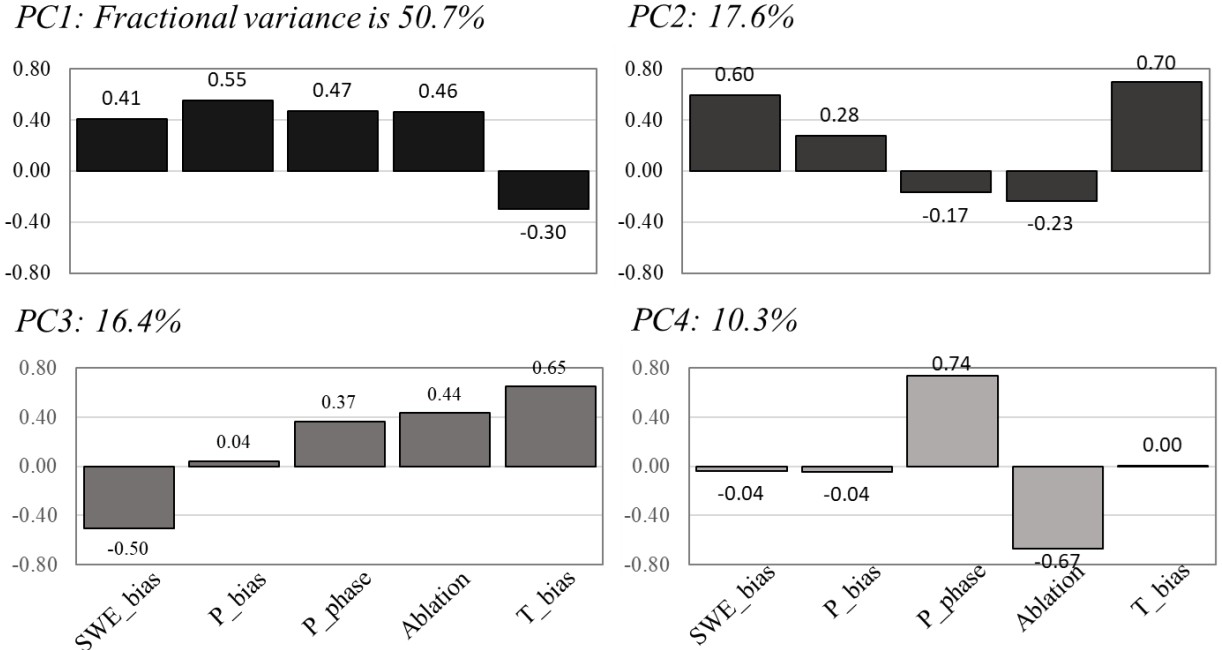

**Figure 8**. Principal component analysis (PCA) on the relations between the SWE uncertainty (*SWE_bias*; values of LSM SWE differences from the observations from **Figure 2**) and the four potential causes (*P_bias*: precipitation bias in **Figure 3**, *P_phase*: precipitation difference between two partitioning methods in **Figure 4**, *T_bias*: mean bias in winter air temperature in **Figure 5**, and *Ablation*: snow ablation difference during the accumulation period as a proxy of melting physics in **Figure 6**)

## 4. Discussion and future perspectives

The results that the LSMs driven by meteorological forcing data from global models and observations systematically underestimate SWE is consistent with and complements the findings of related studies (Pan et al., 2003; Broxton et al., 2016). Pan et al. (2003) evaluated SWE products from four LSMs with a single forcing data from North American Land Data Assimilation System (NLDAS) at approximately 12.5 km spatial resolution. They found that LSMs show systematic bias in the annual maximum SWE when compared to 110 SNOTEL stations with larger differences in the Pacific Northwest and the Sierra Nevada regions. The NLDAS forcing precipitation was consistently lower than the observations by up to 2000 mm annually at certain stations. This corresponds to the widespread underestimates of precipitation up to more than 2000 mm for all meteorological forcings from this study, particularly for the areas with complex terrains. Previous studies have demonstrated uncertainty and challenges in estimating precipitation in gridded datasets over complex terrain (Gutmann et al.,2012; Livneh et al., 2014; Lundquist et al., 2015). Because topography information (e.g., elevation) is used to interpolate gauge-based precipitation and/or aggregate course resolution precipitation datasets to generate higher resolution gridded products, precipitation uncertainties are larger over complex terrain at higher elevations (Henn et al., 2018). Our results provide similar findings that ECMWF and GDAS precipitation biases in **Figure 3** are dependent on elevations (larger differences at higher



elevations), except for MERRA2. The PCA results reveal that the larger precipitation bias is the largest contributor in the first PC (> 50% of total explained variance) to the SWE uncertainties, which is consistent with previous studies (e.g., Raleigh et al., 2015).

In this study, the temperature biases depend on elevation, and the level of the dependence differs by forcing sources (see **Figure 4**). While MERRA2 biases are relatively constant with elevation, GDAS has colder biases with increasing elevation. The results seem to differ from previous findings of Pan et al. (2003) that the winter NLDAS temperature bias was generally constant by elevation. They stated that the constant lapse rate (6.5 °C km$^{-1}$) used to downscale the coarse meteorological fields to a 12.5 km grid is suitable (Cosgrove et al., 2003). However, applying the constant lapse rate (6.5 °C km$^{-1}$) for temperature downscaling may introduce temperature uncertainty. The surface temperature lapse rate varies differently in time and space

by region (Blandford et al., 2008). For example, Minder et al. (2010) revealed that in the Cascade Mountains annual mean lapse rates were 3.9–5.2 °C km$^{-1}$, which is substantially smaller than the 6.5 °C km$^{-1}$ used in this study with substantial geographic differences in lapse rates. Although sensitivity studies with dynamic or static lapse rates were not conducted in this study, it is reasonable to assume that the cold biases in higher elevations found in this study could be partially corrected by applying more appropriate lapse rates regionally. This can result in corrected proportions of precipitation phases as well as the

magnitude and timing of SWE and snowmelt (Lute & Abatzoglou, 2021).

    As shown earlier, the underestimation of SWE is smaller in Noah-MP compared to the other LSMs despite the underestimated forcing precipitation (e.g. ECMWF) in the regions. The smaller underestimates may be partially attributed to the Jordan fractioning method compensating for precipitation error by allowing for snowfall when the air temperature is above 0 °C. The result is that while Noah-MP still underestimates SWE as compared to the SNOTEL observations, it performs better

in this region than the other three LSMs. For other LSMs, a single threshold method for precipitation partitioning largely contributes to the SWE underestimation. This is also supported by examples of time series of accumulated snowfall from LSMs (**Figure S7**). Snowfall estimates of Catchment, JULES, and Noah LSMs with the GDAS forcing are much less than that of Noah-MP with the same forcing (see the snowfall time series in **Figure S7**). In this study, the two precipitation partitioning methods generated differences in annual snowfall by up to 847 mm with an average of 117 mm across the SNOTEL sites in

the western U.S., particularly larger in the Pacific Northwest. However, in the eastern United States, Jordan's method generated too much snowfall and subsequently overestimated SWE as compared to observations from the New York State Mesonet (Letcher et al., 2021). This suggests that applying a certain partitioning method across larger areas (at least the continental United States) can generate spatially different errors. Jennings et al. (2019) found that air temperature partitioning rain and snowfall varies across the Northern Hemisphere, ranging from –0.4 to 2.4 °C (average: 1.0°C) for 95% of the study's

observations. This implies that the higher air temperature threshold (~ 2.5°C) of Jordan's fractional method may overestimate snowfall.

    The two precipitation partitioning approaches used in this study may have limitations. A new precipitation partitioning method incorporating humidity performed better than air temperature-only methods (Jennings et al., 2019). Also, snow simulations were improved when the wet-bulb temperature, defined as the temperature to which air can be cooled to saturation





by the evaporation of water into the air, is used for partitioning precipitation, particularly in the drier continental regions with higher elevation in the western U.S. This is because, as compared to air temperature, the wet-bulb temperature represents the temperature of a falling hydrometeor well (Wang et al., 2019). Given the findings from the above studies, future comparison studies with multiple precipitation partitioning methods considering humidity and/or other meteorological variables in various environments would be helpful to select the best partitioning approach for the land surface and hydrological modeling

communities. Since there are many differences among the LSMs beyond the partitioning scheme, further studies are also required to consider the single and combined impacts of other model physics on the SWE estimates (i.e., albedo and snow-soil interactions).

The SWE underestimates will likely increase if the cold biases in temperature forcings are corrected. This indicates that even though the cold biases may be fortuitously helping to generate larger snowfall instead of rainfall, contributions of the

precipitation underestimation as well as melting physics to the SWE underestimates are still too large to simulate enough SWE. Note that the Colorado Snow Survey Program with the USDA Natural Resources Conservation Service has ongoing work to correct the air temperature for the entire SNOTEL network, beyond the linear equation correction used here and in other studies (Currier et al., 2017; Sun et al., 2019).

It is likely that early-season melting losses are mainly due to melting physics in LSMs (Broxton et al., 2016). In general,

simplified snow layering schemes with a single snow density (e.g., a single snow layer) may cause rapid snowmelt (Suzuki and Zupanski, 2018) because the snow layering schemes influence thermodynamics in the snowpack and the subsequent timing and presence of melt (Dutra et al., 2011). Considering the layering schemes (a single-layer scheme in both JULES and Noah vs. three-layer schemes in both Noah-MP and Catchment), it is perhaps expected that Noah-MP and Catchment showed relatively better spatial agreements with the SNOTEL SWE/total precipitation ratio (**Figure 7**) than Noah and JULES.

A future study investigating other melting physics in LSMs by comparing energy balance components (e.g., net radiation, snow albedo changes, and heat transfer with the ground) might provide a better understanding of the SWE uncertainty issues. Typically, errors in modeled winter albedo were linked to errors in snow-cover fraction (SCF; Roesch, 2006) and tree cover fraction (TCF; Wang et al. 2016). Better quantification of the SCF and/or TCF in LSMs could improve the albedo and consequently snowmelt and SWE simulations. Recent snow model intercomparison exercise results (e.g., ESM-SnowMIP)

provide insights into relationships between surface temperature and SWE estimations (Krinner e al., 2018; Menard et al., 2021). Menard et al. (2021) found clear differences in model ranking between SWE and snow surface temperature (e.g. four of the best models for SWE estimation were among the worst for snow surface temperature). They stated that underestimating surface temperature leads to a colder snowpack that remains on the ground for longer and results in less snowmelt (Conway et al., 2018). While the current results may not be addressed by the surface temperature issue, because our modeled SWE outputs

were generally underestimated despite cold biases in air temperature (Figure 5), it is worth exploring snow and soil temperature variations along with the SWE changes in models with various environments to explicitly understand internal snow processes (Gouttevin et al., 2012).



While the wind-driven processes are typically considered smaller-scale issues, we acknowledge that this may partially influence the current results because the station observations may be affected by wind redistributions and its impact on the

sublimation of snow (Groot Zwaaftink et al., 2013; Hood et al., 1999). This may cause increased biases particularly at higher elevations because regions with higher elevations generally have more complex terrains where blowing/drifting snow frequently occurs due to wind and avalanches (Mott et al., 2018). Most LSMs, including those used in this study, do not have physical processes to simulate preferential deposition and snow redistribution that are crucial in mountain environments (Dadic et al., 2010; Mott et al., 2018; Lehning et al., 2008); further investigations are required to quantify uncertainties driven by these

"missing" physics, along with the continuous efforts to include relevant model physics into LSMs for better SWE simulations.

Because this study relies on the use of point-based SNOTEL measurements, different spatial representativeness between the 5-km gridded outputs and in-situ measurements is an inevitable issue preventing a definitive evaluation of the SWE, precipitation, and temperature simulations. In terms of the physiographic characteristics of SNOTEL sites, the stations tend to be located on flat and sheltered terrain in small forest gaps due to logistical difficulties in steep and densely forested areas

(Meromy et al., 2013; Molotch and Bales, 2006), potentially leading to relatively large snow accumulation as compared to surrounding areas. However, Meromy et al. (2013), using more than 30,000 field observations, found that there were no consistent biases in SNOTEL snow depth values relative to the surrounding mean observed values in California, Colorado, Wyoming, Idaho, and Oregon. Considering that SNOTEL observations have been widely used for snow research, further investigations for quantifying the spatial representativeness of each station with surrounding areas will be helpful.

Regarding the uncertainty in complex terrain, the spatial resolution of the model simulation can add uncertainty to SWE outputs. Despite the relatively high resolution, the 5-km grid of the SEUP ensemble is still too coarse to fully represent local heterogeneity for microphysical features and snow processes in mountainous regions. Within a few square kilometers, many physical controls including terrain, vegetation, and soils play a role in forming spatial heterogeneity in the snowpack (Currier & Lundquist, 2018; Cho et al., 2021; Meromy et al., 2013; Neumann et al., 2006). Lettenmaier et al. (2015) stated that spatial

resolution of snowpack is ideally required to be no coarser than 100 m to characterize mountain snowpack. Also, in regional climate model experiments, high-resolution simulations (4 km or lower) enable microphysical features such as orographic updrafts generating clouds and precipitation to be captured (e.g. Ikeda et al., 2021). This has relevance to multi-resolution snow modeling studies (e.g., Pavelsky et al., 2011; Wrzesien et al., 2017), which showed better agreement between model and point (or reference) SWE when the model was run at a finer resolution (9 and 3 $km^2$). Although the relatively high-resolution

simulations (5-km) in this study may partially alleviate the issue, we cannot fully address the issue. Future studies may focus on accounting for the impact of spatial representativeness on the identified errors in the gridded SWE.

## 5. Conclusions

This study identifies the dominant sources and relative contributions of SWE errors using the SEUP ensemble recently developed by Kim et al. (2021) including four LSMs, Noah-MP, Catchment, JULES, and Noah, with three different

meteorological forcings, ECMWF, GDAS, and MERRA2, across the western United States. There is a widespread





underestimation of SWE from all LSMs up to 1500 mm, although the uncertainty is regionally dependent. Substantial underestimations of precipitation for all meteorological forcings are found to be a primary source of the SWE underestimation particularly in the Pacific Northwest and Southern Rockies with higher elevation ranges (> 3000 m). The precipitation partitioning approach generates different snowfall estimates by up to 800 mm with the same forcing data. In most LSMs, there are large melting losses during the accumulation period, contributing to the underestimation of SWE. Lastly, there are regionally different biases of air temperature up to -3.5 °C in areas with high elevation. Considering the temperature biases contribute to determining the precipitation phase and subsequent uncertainty in snowfall and SWE, particularly in maritime regions, reducing the temperature bias from meteorological forcings and/or an improved temperature downscaling approach (e.g. a time or spatially varying lapse rate) is required to improve the SWE estimation. Among these LSMs, Noah-MP shows the best performance in simulating SWE (less underestimation), likely due to larger snowfall amounts from Jordan's method and better melting physics, even though there are several limitations in the current physics. The results from this study provide insights needed to guide the improvement of LSM's SWE for snow science and climate research. Further studies with sensitivity analysis for each process with relevant parameters in LSMs might be helpful to explicitly quantify their contributions and to ensure that improvements to LSMs in one region do not adversely impact another region.

*Data availability*. The SNOTEL SWE and accumulated precipitation data are available from the U.S. Department of Agriculture Natural Resources Conservation Service National and Climate Center (https://wcc.sc.egov.usda.gov/reportGenerator/). The bias-corrected air temperature data for the SNOTEL sites are available from Pacific Northwest National Laboratory at https://www.pnnl.gov/data-products. Time series outputs from the SEUP ensemble corresponding to the 809 stations are available used in this paper are available for download at [*will add a link to data from Hydroshare, currently being setup with an ODC Attribution (ODC-BY) license for access without restrictions*]. The MERRA2 forcing dataset is distributed by the NASA Goddard Global Modeling and Assimilation Office (GMAO; https://gmao.gsfc.nasa.gov/reanalysis/MERRA-2/data_access/). The GDAS forcing data are publicly available from the US National Centers for Environmental Prediction (NCEP; https://nomads.ncep.noaa.gov/pub/data/nccf/com/gfs/prod). The ECMWF forcing data are not publicly available but made available under license (https://www.ecmwf.int/en/forecasts/datasets).

*Author contributions*. EC led the investigation, conceptualized the research, did the formal analysis, and wrote the initial draft. CMV and SVK conceptualized the research, took responsibility for the investigation, acquired the funding and the resources, supervised the project, and reviewed and edited the paper. RSK did the model simulations, helped with the investigation, and reviewed and edited the paper. MLW and JMJ helped with the investigation, provided technical and scientific inputs, and reviewed and edited the paper.

*Competing interests*. On behalf of all authors, the corresponding author states that there is no conflict of interest




*Acknowledgments*. The authors are grateful to all colleagues who contributed to the SEUP project. This research gratefully acknowledges support from NASA Terrestrial Hydrology Program (NNH16ZDA001N). Computing resources to run the NASA land information system (LIS) were supported by the NASA Center for Climate Simulation.

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
