# Peer review of "Precipitation Biases and Snow Physics Limitations Drive the Uncertainties in Macroscale Modeled Snow Water Equivalent"

_Hydrology and Earth System Sciences, 2022_

## Author Comment (AC3)

**RC2: 'Comment on hess-2022-136', Anonymous Referee #2, 24 Jun 2022**

Cho et al. use a 12-member ensemble from 4 land surface models (LSMs) and 3 sets of meteorological forcings to assess uncertainties in snow water equivalent (SWE) estimates. Using principal component analysis (PCA), they try to identify the source of error in the SWE estimates and consequently attribute the uncertainty in SWE to various factors such as precipitation bias, etc.

The paper is well-written and easy to read. Results from this paper can guide advancements in land surface modeling of SWE, which is important since errors in SWE can translate into errors in other hydrologic variables such as runoff and soil moisture. The paper highlights the extent that biases in precipitation can contribute to uncertainty in SWE as well as other factors.

[Answer] Thank you to the Reviewer for the positive feedback and constructive comments which helped us to improve the manuscript. Please see our response to each comment given below.

**General Comments:**

It is important to acknowledge that there are uncertainties in the SNOTEL measurements early on in the paper. Also, SNOTEL measurements are not necessarily representative of the surrounding regions. Please clarify this as it relates to L.149, which seems to contradict L.341-349, where the authors do acknowledge that in-situ measurements may not always be representative of surrounding areas.

[Answer] Thank you for pointing out it. This is reasonable. To address this, we removed L.149's statement in Section 2.2 SNOTEL data and added relevant statements earlier in the manuscript (Introduction) that limited spatial representativeness of some SNOTEL stations could lead to uncertainties in the comparison below.

"This study seeks to identify the primary sources of the errors and to quantify their contributions to the modeled SWE uncertainty (forcing errors vs. snow-related physics) during the accumulation periods for eight years (2010 to 2017) against the 809 Snowpack Telemetry (SNOTEL) stations. We acknowledge that some SNOTEL stations are not spatially representative of surrounding areas (Meromy et al., 2013; Molotch & Bales, 2006), potentially leading to uncertainties in the comparison. Nevertheless, we assume that more than 800 SNOTEL sites across the western U.S. would be sufficient to quantify macroscale LSM SWE uncertainties."

- Meromy, L., Molotch, N. P., Link, T. E., Fassnacht, S. R., & Rice, R.: Subgrid variability of snow water equivalent at operational snow stations in the western USA. Hydrological Processes, 27(17), 2383-2400, 2013.
- Molotch, N. P., & Bales, R. C.: SNOTEL representativeness in the Rio Grande headwaters on the basis of physiographics and remotely sensed snow cover persistence. Hydrological Processes: An International Journal, 20(4), 723-739, 2006.

Several of the findings appear to be consistent with previous work. It is not always clear what additional findings beyond those consistencies would add to the existing body of literature. Include additional discussions that clearly show the value of this work instead of simply confirming what others have previously found.

[Answer] Thank you for the suggestion. In response to the suggestion, we included summarized statements addressing the value of this work beyond the previous findings from the existing body of literature.

"Overall, the primary value of this study is to provide a big picture of LSM-based SWE and error sources of the uncertainty derived from meteorological forcings and limited physical processes of the LSMs using a large ensemble with four LSMs and three meteorological forcings. As compared to previous studies, this work not only identified primary sources of uncertainty but also quantified the relative contributions of the error sources on the uncertainty of macroscale modeled SWE. Thus, this study helps comprehensively understand current state of SWE performances from well-known LSMs and reanalysis products and provides insights into future improvement of snow modeling in LSM."

It would be valuable to consider biases in timing as it could introduce uncertainty into the season totals and subseasonal dynamics of the snowpack. Has the timing of melt events been carefully evaluated as event timing impacts the total accumulation of SWE?

[Answer] Thank you for the question. We agreed with the reviewer's thought that the biases in timing of melt events could be valuable information to better understand the uncertainty into the total accumulation of SWE. As shown in Figure 1b, we investigated timings of the maximum total accumulation of SWE which can be a proxy of the timing of melt events as melt events occur right after the peak SWE. The mean timings of the peak SWE are on average 36 days earlier than the observation, indicating earlier melting events (right after the peak SWE) may be related to less accumulation of SWE. We also found that the patterns of the earlier timings are more apparent in areas with higher elevation ranges than with lower elevation. Although we did not specifically focus on the current study how the timings and frequencies of the "intermittent" melt events during the accumulation period impact the seasonal dynamics of the snowpack and how the models simulate those melt events compared to observations would be valuable which can be considered as a future study.

There are a few different analysis windows that are referred to as winter in the paper. For instance, 1st October to 31st May is used in some cases; however, winter accumulated precipitation is cumulative precipitation from October 1st through the date of the maximum SWE. It is unclear why a consistent time period is not used. Also, different terminology should be employed to differentiate these various "winter" periods, especially since 1st October to 31st May includes other seasons.

[Answer] Thank you for pointing out this. As we stated earlier in this manuscript (Lines 88-90), we focused on the snow accumulation period which is defined as October 1st to the date of the annual maximum SWE of each station throughout the manuscript, except only for comparing mean temperature differences between three forcings and bias-corrected SNOTEL data which

was used 1st October to 31st May. Considering the onset of above-freezing temperatures generally occurs between March and May in maritime and continental regions over the West (Trujillo & Molotch, 2014), we assumed this period is generally acceptable to check air temperature biases from meteorological forcings during the winter season. To address the reviewer's point, we modified and added statements.

"This study seeks to identify the primary sources of the errors and to quantify their contributions to the modeled SWE uncertainty (forcing errors vs. snow-related physics) during the accumulation periods for eight years (2010 to 2017) against the 809 Snowpack Telemetry (SNOTEL) stations. We focus on the snow accumulation period which is defined as October 1st to the date of the annual maximum SWE of each station. For winter temperature comparison only, a period from 1st October to 31st May is used by assuming a typical winter period (Trujillo & Molotch, 2014)."

Trujillo, E., & Molotch, N. P. (2014). Snowpack regimes of the western United States. Water Resources Research, 50(7), 5611-5623. https://doi.org/10.1002/2013WR014753

**Specific Comments:**

Section 3.1/Figure 1. It seems that difference maps would be more powerful for conveying the uncertainty across LSMs relative to SNOTEL than the current maximum SWE maps and dates. Consider including difference maps in the main text with the SNOTEL column from the current Figure 1 and moving the other 4 columns to the supplement to show the actual magnitudes and dates.

[Answer] Thank you for your good suggestion. We've modified and merged Figures 1 & 2 including difference maps for mean annual maximum SWE and dates. The actual SWE magnitudes and dates of LSMs were moved to Supplementary Material (Figure S2).

[Figure]

**Figure 1**. (a) Mean annual maximum SWE map of SNOTEL observations and SWE difference maps of the four land surface model (LSM)'s annual maximum values from the SNOTEL observations with density plots of the SWE difference for each LSM by four elevation ranges of SNOTEL sites (a solid black vertical line in the density plots represents median value) and (b) mean annual maximum SWE date map of SNOTEL and date difference maps along density plots when the annual maximum SWE for each LSM occurs from 2010 to 2017 across the western United States. Each LSM results are averaged over the three different forcings. Elevation map with the four ranges is provided in Figure S4.

[Figure]

**Figure S2**. (a) Mean annual maximum SWE maps of SNOTEL observations and the four LSMs averaged over the three forcings and (b) mean date maps when the annual maximum SWE occurs from 2010 to 2017 across the western

Figure 2b. In the Figure 2b caption, mention what the vertical lines represent.

[Answer] Thank you for pointing out. We included the meaning of the black vertical lines which represents "median" values of SWE difference for the four elevation groups.

"Figure 1. (a) Mean annual maximum SWE map of SNOTEL observations and SWE difference maps of the four land surface model (LSM)'s annual maximum values from the SNOTEL observations with density plots of the SWE difference for each LSM by four elevation ranges of SNOTEL sites (**a solid black vertical line in the density plots represents median value**) and (b) mean annual maximum SWE date map of SNOTEL and date difference maps along density plots when the annual maximum SWE for each LSM occurs from 2010 to 2017 across the western United States. Each LSM results are averaged over the three different forcings. Elevation map with the four ranges is provided in Figure S4."

Section 3.3/Figure 4. Include an equivalent map for the SNOTEL snowfall and how the snowfall is distributed across elevations. Difference maps relative to SNOTEL would be useful.

[Answer] Thank you for the suggestion. Unfortunately, we cannot include an equivalent map for SNOTEL snowfall, because SNOTEL snowfall data are not available (total precipitation was available only).

L.271: Remove language such as "seem" here. In an effort to avoid any potential confusion since NLDAS is not used by Cho et al., it may be better to rephrase as "Our results differ from Pan et al. (2003) who concluded that the winter temperature bias was generally constant with elevation when using NLDAS."

[Answer] Thank you for pointing out. That makes sense to us. We've modified the statement as you suggested.

"The results differ from previous findings of Pan et al. (2003) who concluded that the winter NLDAS temperature bias was generally constant by elevation."

L.359: Should l.359 read 9 and 3 "km" as opposed to "km2" since Pavelsky et al. (2011) and Wrzesien et al. (2017) use resolutions of 27 km, 9 km, and 3 km?

[Answer] Yes, you are correct. We edited this to "km".

---

## Author Response (AR1)

**Editor's comments to the author:**

Thanks for submitting this interesting manuscript to HESS. Both reviewers found the manuscript publishable with only minor revisions and your responses in the discussion phase indicate that you will be able to utilize these valuable suggestions to improve your work further.

Reading your manuscript, I saw links to some recent work in our group, see for instance references below. No need at all to include/cite these (especially as one is in German), but I thought this could be of interest for you.

Best regards, Jan Seibert

Lopez, M.G., Vis, M.J.P., Jenicek, M., Griessinger, N., Seibert, J., 2020. Assessing the degree of detail of temperature-based snow routines for runoff modelling in mountainous areas in central Europe. Hydrol. Earth Syst. Sci. 24. https://doi.org/10.5194/hess-24-4441-2020

Freudiger, D., Frielingsdorf, B., Stahl, K., Steinbrich, A., Weiler, M., Griessinger, N., & Seibert, J. (2016). Das Potential meteorologischer Rasterdatensätze für die Modellierung der Schneedecke alpiner Einzugsgebiete. Hydrologie und Wasserbewirtschaftung, 60(6), 353-367. https://doi.org/10.5675/HyWa\_2016,6\_1

**Dear Dr. Jan Seibert,**

Thank you for letting us know about your group's work which are very interesting providing an impact of the temperature-index snow routines on runoff estimations in a mountain environment. I believe the studies would be greatly helpful for our future research on investigating the effect of the SWE uncertainties on runoff simulations. Also, we would like to thank you and the two reviewers for taking their time to provide constructive comments that improved this paper.

Sincerely, Eunsang et al.

**CC1: 'Comment on hess-2022-136', Ross Brown, 17 May 2022**

Citation: https://doi.org/10.5194/hess-2022-136-CC1

Brown et al (2018) demonstrated the same issues with precipitation-related errors in simulated SWE over southern Quebec. See https://doi.org/10.1002/hyp.13221

[Answer] Thank you, Ross Brown, for letting us know about your great work which addressed the issues with solid precipitation-driven errors in SWE simulations in southern Québec. We included additional discussions in the Discussion part.

"Brown et al. (2018) also demonstrated insufficient solid precipitation from gridded reanalysis and model products led to systematic errors in SWE estimations in southern Québec, indicating the need for improving estimates of snowfall. These results correspond to the widespread underestimates of precipitation up to more than 2000 mm for all meteorological forcings from this study, particularly for the mountainous areas with complex terrains."

Brown, R., Tapsoba, D., & Derksen, C. (2018). Evaluation of snow water equivalent datasets over the Saint-Maurice river basin region of southern Québec. Hydrological Processes, 32(17), 2748-2764.

**RC1: 'Comment on hess-2022-136', Anonymous Referee #1, 17 Jun 2022**

General comments:

The manuscript by Eunsang Cho and others is well organized and clearly presented. The research fits well into the larger picture of mountain snow research and highlights the need for improving LSM estimates of SWE. The authors imply a focus on precipitation/snowfall accumulation is an important first step. Without proper precipitation accumulations, the model is unable to properly evolve the snowpack. It is important to identify the issues with LSM SWE estimates and this manuscript does just that. It does not rank the LSM outputs, but rather uses them to provide strong conclusions about the next steps in improving the models.

The authors provide a lengthy discussion that addresses the main shortcomings of the models and observations used in their research. This provides good context to how their work fits into the larger picture of snow research and I found the discussion to be just as important as the rest of the paper.

I am happy to have reviewed this paper and know of the conclusions. The paper receives excellent marks in terms of the HESS review criteria of scientific significance, scientific quality, and presentation quality. Thus, I recommend this paper be accepted to HESS. I have given a few minor suggestions below that may contribute to the improvement of the manuscript:

**[Answer] Thank you to the Reviewer for the positive feedback and constructive comments which helped us to improve the manuscript. Please see our response to each comment given below.**

1) When discussing the potential of using wet-bulb temperature as a rain/snow partitioning method, the inclusion of Sims and Liu, 2015 (https://journals.ametsoc.org/view/journals/hydr/16/4/jhm-d-14-0211\_1.xml) would be beneficial to the reader. This partitioning method is used for satellite remote sensing of precipitation.

[Answer] Thank you for the important literature. We have added the recommended literature in the section "4. Discussion and future perspectives" where we discussed the potential of using wet-bulb temperature as a rain/snow partitioning method as below.

"The two precipitation partitioning approaches used in this study may have limitations. A new precipitation partitioning method incorporating humidity performed better than air temperatureonly methods (Jennings et al., 2019). Also, solid precipitation simulations were improved when the wet-bulb temperature, defined as the temperature to which air can be cooled to saturation by the evaporation of water into the air, was used, particularly in the drier, high elevation continental regions of the western U.S. This was because, as compared to air temperature, the wet-bulb temperature was closer to the actual temperature of a falling hydrometeor (Sims and Liu, 2015; Wang et al., 2019). Considering that the wet-bulb temperature is affected by surface skin temperature and vertical lapse rate (Sims and Liu, 2015), future comparison studies with multiple precipitation partitioning methods should consider humidity, wet-bulb temperature, and/or other meteorological variables in various environments in developing the best partitioning approach for the land surface and hydrological modeling communities."

Sims, E. M., & Liu, G. (2015). A parameterization of the probability of snow-rain transition. Journal of hydrometeorology, 16(4), 1466-1477.

**2) A few minor corrections:**

Line 74: Rephrase "Furthermore, most of the prior studies used a single or multiple LSMs with a single meteorological forcing and/or simulated/reanalysis SWE with relatively coarse spatial resolutions (e.g., 12.5 km to 50 km), which impedes the quantification of the contributions by producing additional uncertainties." ---> "Furthermore, most of the prior studies used a single or multiple LSMs with one meteorological forcing and either simulated or reanalysis SWE with relatively coarse spatial resolutions...."

[Answer] Thank you for rephrasing the sentence which makes more sense. We applied this as below.

"Furthermore, most of the prior studies used a single or multiple LSMs with one meteorological forcing and either simulated or reanalysis SWE with relatively coarse spatial resolutions (e.g., 12.5 km to 50 km), which impedes the quantification of the contributions by producing additional uncertainties."

Line 163: Simplify "The data matrix was pre-processed: the values in each column were normalized with the following two steps: 1) the mean of each column is zero, and 2) each column was standardized to the unit norm as the variables have different units." ---> "Data in the matrix was pre-processed such that the mean and standard deviation of each variable is zero and one, respectively."

[Answer] We appreciate your suggestion. We applied the simplified statement in the manuscript.

"The potential sources of the error are obtained from the comparison between SEUP and SNOTEL observations. Data in the matrix was pre-processed such that the mean and standard deviation of each variable is zero and one, respectively."

Line 199: Change "... fractioning method partitions partial precipitation..." ---> "... fractioning method partitions precipitation..."

[Answer] Thank you for the correction. We agreed and applied this.

"This is not surprising because the fractioning method partitions precipitation amounts with air temperatures ranging from 0 to 2.5  $^{\circ}$ C as snowfall, which would be classified as liquid rainfall with a single threshold method that uses 0  $^{\circ}$ C as the rain-snow threshold."

**RC2: 'Comment on hess-2022-136', Anonymous Referee #2, 24 Jun 2022**

Cho et al. use a 12-member ensemble from 4 land surface models (LSMs) and 3 sets of meteorological forcings to assess uncertainties in snow water equivalent (SWE) estimates. Using principal component analysis (PCA), they try to identify the source of error in the SWE estimates and consequently attribute the uncertainty in SWE to various factors such as precipitation bias, etc.

The paper is well-written and easy to read. Results from this paper can guide advancements in land surface modeling of SWE, which is important since errors in SWE can translate into errors in other hydrologic variables such as runoff and soil moisture. The paper highlights the extent that biases in precipitation can contribute to uncertainty in SWE as well as other factors.

[Answer] Thank you to the Reviewer for the positive feedback and constructive comments which helped us to improve the manuscript. Please see our response to each comment given below.

**General Comments:**

It is important to acknowledge that there are uncertainties in the SNOTEL measurements early on in the paper. Also, SNOTEL measurements are not necessarily representative of the surrounding regions. Please clarify this as it relates to L.149, which seems to contradict L.341-349, where the authors do acknowledge that in-situ measurements may not always be representative of surrounding areas.

[Answer] Thank you for pointing out it. This is reasonable. To address this, we removed L.149's statement in Section 2.2 SNOTEL data and added relevant statements earlier in the manuscript (Introduction) that limited spatial representativeness of some SNOTEL stations could lead to uncertainties in the comparison below.

"This study seeks to identify the primary sources of the errors and to quantify their contributions to the modeled SWE uncertainty (forcing errors vs. snow-related physics) during the accumulation periods for eight years (2010 to 2017) against the 809 Snowpack Telemetry (SNOTEL) stations. We acknowledge that some SNOTEL stations are not spatially representative of surrounding areas (Meromy et al., 2013; Molotch & Bales, 2006), potentially leading to uncertainties in the comparison. Nevertheless, we assume that more than 800 SNOTEL sites across the western U.S. would be sufficient to quantify macroscale LSM SWE uncertainties."

- Meromy, L., Molotch, N. P., Link, T. E., Fassnacht, S. R., & Rice, R.: Subgrid variability of snow water equivalent at operational snow stations in the western USA. Hydrological Processes, 27(17), 2383-2400, 2013.
- Molotch, N. P., & Bales, R. C.: SNOTEL representativeness in the Rio Grande headwaters on the basis of physiographics and remotely sensed snow cover persistence. Hydrological Processes: An International Journal, 20(4), 723-739, 2006.

Several of the findings appear to be consistent with previous work. It is not always clear what additional findings beyond those consistencies would add to the existing body of literature. Include additional discussions that clearly show the value of this work instead of simply confirming what others have previously found.

[Answer] Thank you for the suggestion. In response to the suggestion, we included summarized statements addressing the value of this work beyond the previous findings from the existing body of literature.

"Overall, the primary value of this study is to provide a big picture of LSM-based SWE and error sources of the uncertainty derived from meteorological forcings and limited physical processes of the LSMs using a large ensemble with four LSMs and three meteorological forcings. As compared to previous studies, this work not only identified primary sources of uncertainty but also quantified the relative contributions of the error sources on the uncertainty of macroscale modeled SWE. Thus, this study helps comprehensively understand current state of SWE performances from well-known LSMs and reanalysis products and provides insights into future improvement of snow modeling in LSM."

It would be valuable to consider biases in timing as it could introduce uncertainty into the season totals and subseasonal dynamics of the snowpack. Has the timing of melt events been carefully evaluated as event timing impacts the total accumulation of SWE?

[Answer] Thank you for the question. We agreed with the reviewer's thought that the biases in timing of melt events could be valuable information to better understand the uncertainty into the total accumulation of SWE. As shown in Figure 1b, we investigated timings of the maximum total accumulation of SWE which can be a proxy of the timing of melt events as melt events occur right after the peak SWE. The mean timings of the peak SWE are on average 36 days earlier than the observation, indicating earlier melting events (right after the peak SWE) may be related to less accumulation of SWE. We also found that the patterns of the earlier timings are more apparent in areas with higher elevation ranges than with lower elevation. Although we did not specifically focus on the current study how the timings and frequencies of the snowpack and how the models simulate those melt events compared to observations would be valuable which can be considered as a future study.

There are a few different analysis windows that are referred to as winter in the paper. For instance, 1st October to 31st May is used in some cases; however, winter accumulated precipitation is cumulative precipitation from October 1st through the date of the maximum SWE. It is unclear why a consistent time period is not used. Also, different terminology should

be employed to differentiate these various "winter" periods, especially since 1st October to 31st May includes other seasons.

[Answer] Thank you for pointing out this. As we stated earlier in this manuscript (Lines 88-90), we focused on the snow accumulation period which is defined as October 1st to the date of the annual maximum SWE of each station throughout the manuscript, except only for comparing mean temperature differences between three forcings and bias-corrected SNOTEL data which was used 1st October to 31st May. Considering the onset of above-freezing temperatures generally occurs between March and May in maritime and continental regions over the West (Trujillo & Molotch, 2014), we assumed this period is generally acceptable to check air temperature biases from meteorological forcings during the winter season. To address the reviewer's point, we modified and added statements.

"This study seeks to identify the primary sources of the errors and to quantify their contributions to the modeled SWE uncertainty (forcing errors vs. snow-related physics) during the accumulation periods for eight years (2010 to 2017) against the 809 Snowpack Telemetry (SNOTEL) stations. We focus on the snow accumulation period which is defined as October 1st to the date of the annual maximum SWE of each station. For winter temperature comparison only, a period from 1st October to 31st May is used by assuming a typical winter period (Trujillo & Molotch, 2014)."

Trujillo, E., & Molotch, N. P. (2014). Snowpack regimes of the western United States. Water Resources Research, 50(7), 5611-5623. https://doi.org/10.1002/2013WR014753

**Specific Comments:**

Section 3.1/Figure 1. It seems that difference maps would be more powerful for conveying the uncertainty across LSMs relative to SNOTEL than the current maximum SWE maps and dates. Consider including difference maps in the main text with the SNOTEL column from the current Figure 1 and moving the other 4 columns to the supplement to show the actual magnitudes and dates.

[Answer] Thank you for your good suggestion. We've modified and merged Figures 1 & 2 including difference maps for mean annual maximum SWE and dates. The actual SWE magnitudes and dates of LSMs were moved to Supplementary Material (Figure S2).